# Safety and Efficacy of Several Versus Isolated Prophylactic Flexor Tenotomies in Diabetes Patients: A 1-Year Prospective Study

**DOI:** 10.3390/jcm11144093

**Published:** 2022-07-14

**Authors:** Mateo López-Moral, Raúl J. Molines-Barroso, Yolanda García-Álvarez, Irene Sanz-Corbalán, Aroa Tardáguila-García, José Luis Lázaro-Martínez

**Affiliations:** 1Diabetic Foot Unit, Facultad de Enfermería, Fisioterapia y Podología, Universidad Complutense de Madrid, 28040 Madrid, Spain; matlopez@ucm.es (M.L.-M.); ygarci01@ucm.es (Y.G.-Á.); irsanz01@ucm.es (I.S.-C.); aroa.tardaguila@ucm.es (A.T.-G.); diabetes@ucm.es (J.L.L.-M.); 2Instituto de Investigación Sanitaria del Hospital Clínico San Carlos (IdISSC), 28040 Madrid, Spain

**Keywords:** diabetic foot, surgery, deformity, prevention

## Abstract

Background: To assess long-term clinical outcomes of patients who underwent isolated versus several percutaneous flexor tenotomies for the treatment of toe deformities and previous diabetic foot ulcers; Methods: Twenty-three patients (mean age 66.26 ± 11.20, years) who underwent prophylactic percutaneous flexor tenotomies secondary to tip-toe ulcers participated in this 1-year prospective study. The study was stratified into two groups for analyses: (1) isolated tenotomies patients, and (2) several tenotomies patients (two or more tenotomies). Outcome measures were toe reulceration and recurrence, minor lesions, digital deformities, and peak plantar pressure (PPP—N/cm^2^) and pressure/time Integral (PTI—N/cm^2^/s) in the hallux and minor toes after a 1-year follow-up period; Results: Patients with isolated tenotomies (*n* = 11, 35.48%) showed a higher rate of reulceration (*n* = 8, 72.7%, *p* < 0.001) in the adjacent toes, additionally, we found more prevalence of hyperkeratosis (*n* = 11, 100%), minor lesions (*n* = 9, 81%), and claw toes (*n* = 11, 100%) (*p* < 0.001). In several tenotomies patients (*n* = 20, 64.52%), we found a higher rate of floating toes (*n* = 16, 80%) in comparison with isolated tenotomies patients (*p* < 0.001). PPP and PTI in the non-tenotomy toes were higher in the group of patients who underwent isolated tenotomies (*p* < 0.001); Conclusions: Patients who underwent several tenotomies had better clinical outcomes after a 1-year follow-up period compared to isolated tenotomies.

## 1. Introduction

Diabetic foot ulcers (DFU) on the dorsal and plantar aspects of the toes have become one of the most common locations of ulcer occurrence in the foot [1], reaching a total prevalence of foot ulceration that ranges from 43% to 55% [2]. Although toe ulcers are smaller than the metatarsal head, midfoot, or rearfoot ulcers, they are misdiagnosed and tend to have higher rates of foot amputation compared to other locations of the foot [3].

Motor neuropathy leads to atrophy of intrinsic foot muscles (interossei and lumbricals); when the intrinsic muscles become atrophic and overpowered by the extrinsic muscles, the stabilizing action is lost, which may eventually result in claw or hammer toes [4]. Those toe deformities have been linked to callus formation secondary to higher plantar pressures [5]. In persons with diabetes and neuropathy, the toe deformity increases plantar pressures during midstance and toe-off, which can result in abundant callus formation, minor lesions, and ultimately toe ulceration in the tip of the toes [6]. Off-loading and debridement of tip-toe ulcers are the basis of the treatment to achieve ulcer healing; despite this, conservative treatment via orthotic interventions, such as therapeutic footwear, toe spacers, or padding, remains unclear and has a weak level of evidence [7].

Flexor tendon tenotomies have been demonstrated to be an effective, safe, and easy procedure that is advocated for flexible toe deformities and can be performed prophylactically or curatively to alleviate the focal pressure on ulcerated areas [8]. The International Working Group Diabetic Foot (IWGDF) recommends performing digital flexor tendon tenotomies in a person with diabetes and abundant callus or an ulcer on the apex or distal part of a non-rigid hammer toe to prevent the first ulcer or the development of a recurrent foot ulcer [9]. Digital flexor tendon tenotomies have been demonstrated to reduce the risk of recurrences in the tip of the toes relative to other non-surgical techniques [10,11,12]. A previous systematic review reported 250 flexor tenotomy procedures performed in 163 patients [13]; the included studies generally reported good healing rates (92–100% within two months post-operative follow-up), and relatively few recurrences during a middle-term follow-up (0–18% at 22 months median post-operative follow-up). Conservative surgery procedures, such as metatarsal head resection, Achilles tendon lengthening, or joint arthroplasty resection, have shown high reulceration rates, indicating that prophylactic surgery could reduce the safety of this procedure [14]. Several authors reported transfer ulcers on adjacent toes due to transferred pressure [11,12,15].

Flexor tendon tenotomies are widely used to treat and prevent toe ulcers, despite this, no previous research has analyzed outcomes after percutaneous flexor tendon tenotomies and if there exists any relationship between patients with several versus isolated tenotomies. We hypothesize that implementing several percutaneous flexor tenotomies in patients with previous tip-toe ulcers can reduce the reulceration rate in comparison with isolated percutaneous flexor tenotomies.

Therefore, the principal aim of this study was to assess the long-term clinical outcomes of patients who underwent isolated versus several percutaneous flexor tenotomies for the treatment of toe deformities and previous diabetic foot ulcers.

## 2. Materials and Methods

### 2.1. Subjects

Twenty-three patients who underwent percutaneous flexor tenotomies secondary to tip-toe ulcers participated in this 1-year prospective study in a specialized diabetic foot unit between April 2019 and January 2021. All the patients included in the study suffered from a previous orthopedic treatment failure, defined as ulcer recurrence in the same toe by using toe spacers and therapeutic footwear modifications [9].

The inclusion criteria were confirmed type 1 or type 2 diabetes, age > 18 years, affected with flexible toe deformities, history of tip-toe DFU, and loss of protective foot sensation because of peripheral neuropathy (DPN).

Exclusion criteria were ulcers during the examination, transmetatarsal or major amputation in the contralateral limb (below or above the knee), history of rheumatoid disease, other causes of neuropathy, critical limb ischemia as defined according to the IWGDF guidance [16], and the need for walking aids. Patients with previous toe surgery were also excluded (both musculoskeletal and soft tissue procedures).

After institutional review, board approval was obtained, and patients’ medical records and clinicopathologic conditions were recorded. Ethical approval was obtained (19/173-E) on 30 April 2019, and the study was completed following the ethical standards of the responsible committee. Informed consent was obtained from each patient. The authors declare that they complied with the code of ethics of the Declaration of Helsinki [17].

### 2.2. Clinical Evaluation

At baseline, clinical characteristics were collected in the case report form (CRF) after the patient signed informed consent on day zero. Body mass index (BMI) was calculated as weight (kg) divided by height (m^2^). Clinicopathologic data, including diabetes type, hypertension, and HbA1c (%) values in the previous 3 months were collected. The patients’ renal, cardiac, and retinopathy status and previous minor amputation were recorded in the clinical record form from the patient’s clinical history. DPN was diagnosed according to the inability to sense the pressure of a 10-g Semmes-Weinstein monofilament at three plantar foot sites and/or a vibration perception threshold >25 V as assessed using a biothesiometer (Me.Te.Da. s.r.l., Via Silvio Pellico, 4, 63074 San Benedetto del Tronto, Italy) [18]. According to the IWGDF guidelines, critical limb ischemia is defined as the absence of both distal pulses and a brachial ankle index of <0.39, systolic ankle pressure <50 mmHg, and toe pressure <30 mmHg [19].

### 2.3. Percutaneous Flexor Tendon Tenotomy Procedure

Patients were subject to a biomechanical assessment. First, the flexibility of the deformity was evaluated using the Kellikian push-up test [20]. If the deformity was corrected by applying pressure at the base of the metatarsophalangeal joint (MTPJ), it was then considered flexible; in contrast, the persistence of clawing or hammer indicated a rigid deformity [20,21]. Patients with flexor tenotomy indications were those patients with previous DFU on the tip of the toe. No patient with curative indication was included in the study [22]. The procedure consisted of locating the flexor tendon by placing it under tension followed by a subsequent transversal incision in the flexor digitorium longus and brevis (both for minor toes and hallux) in the proximal portion of the proximal phalanx (Figure 1). The wound following the tenotomy was sutured [14]. All the procedures were performed by the same senior surgeon (JLLM) in the same operating room. The patients received postoperative antibiotic prophylaxis with amoxicillin/clavulanic acid 875/125 mg, 1 g every 8 h, starting just after the procedure and continuing for a week. The patient was asked to walk with a post-op extra depth shoe for 7 days. The suture was removed after 7 days, and then the patient was asked to wear extra-depth therapeutic footwear with a fully customized insole to decrease peak pressures in the plantar aspect of the foot [23].

Patients were subjected to isolated tenotomies when only one toe was affected with a previous toe ulcer, several tenotomies were performed in patients with one or more toes affected with previous toe ulcers, and in addition, any of the remaining toes were affected with a flexible toe deformity.

### 2.4. Plantar Pressure Measurement

A dynamic pressure measurement system (Footscan^®^ system, RSscan International, 3583 Olen, Belgium) was used to record the peak plantar pressure (PPP) (N/cm^2^) and pressure-time integral (PTI) (N/cm^2^/s) in the dynamic barefoot condition after a 1-year follow-up period. The hardware included a 2-m plate with four sensors/cm^2^ and a 3D-Box interface that was synchronized with a motion capture system. All data were recorded at a measurement frequency of 500 Hz and were processed using Scientific Footscan^®^ software (RSscan International, 3583 Olen, Belgium). To accommodate the patient to the normal gait and speed, patients were instructed to walk barefoot for 3 min before measuring the plantar pressure. After this, four registers were taken to calculate the mean of both measurements (PPP and PTI) with a two-step approach to the platform [24]. The software divides every foot into ten sectors: the hallux, toes, first to fifth metatarsal head, midfoot, medial heel, and lateral heel. For the analyses, only the hallux and minor toes were extracted and evaluated to assess changes in barefoot pressure after tenotomy procedures. The investigator who analyzed and extracted data from PPP and PTI parameters was blinded to the clinical data from every patient; both investigators are experienced podiatrists with more than 5 years of experience in the treatment of diabetic foot complications. All clinical evaluations were carried out in a specialized biomechanics laboratory.

### 2.5. Follow-Up

All patients were followed up for 1 year. Patients came monthly to the outpatient clinic, according to the international guidelines [9]. At every visit, the principal investigator performed debridement of high-risk points, such as minor lesions or hyperkeratosis.

### 2.6. Outcome Measures

The main outcome measure was to assess the presence of percutaneous flexor tenotomies clinical outcomes after a 1-year follow-up period. Toe reulceration and recurrence were defined according to the IWGDF guidelines as a break of the skin of the foot that involves, as a minimum, the epidermis and part of the dermis, in a different toe than the previous toe and the same toe, respectively [25].

Minor lesions were defined as nonulcerative lesions of the skin on the plantar aspect of the tip of the toes and included abundant callus, hemorrhage, or a blister [26]. The following digital deformities were assessed: Claw toes were considered only when the proximal phalanx was extended. Floating toes were defined when all the metatarsophalangeal and interphalangeal joints were in hyperextension position, and the tip of the toe did not contact the floor [27] (Figure 1).

The secondary outcome measure was to evaluate the differences between groups in barefoot pressures after the flexor tenotomy procedure in the hallux and the minor toes.

The investigator who assessed DFU ulceration and foot biomechanics was blinded to the percutaneous flexor tenotomy procedure to avoid bias in outcome interpretation.

### 2.7. Statistical Analyses

The assumption of the normality of all continuous variables was verified using the Kolmogorov-Smirnov test. Quantitative variables were presented as the mean and standard deviation (SD), while qualitative variables were presented as percentages and frequencies. To explore differences in clinical features between patients with several versus isolated tenotomies, the Chi-square test for categorical variables and the Student’s *t*-test for quantitative variables were performed. *p*-values < 0.05 were considered statistically significant, with confidence intervals of 95%. All statistical analyses were performed using SPSS statistics version 25.0 for Mac OS (SPSS, Chicago, IL, USA). The study was stratified into two groups for analyses: (1) isolated tenotomies patients, and (2) several tenotomies patients (two or more tenotomies). A secondary multiple comparison sub analysis (Bonferroni test) was performed between the subgroups of tenotomy patients (one tenotomy, three tenotomies, four tenotomies, and five tenotomies) for the primary outcome measure.

The sample size calculation based on a 28-month follow-up study of patients with previous flexor tenotomy of the flexor digitorium longus to heal tip toe ulcers [10] showed a recurrence rate of 12.1%. As a relevant risk reduction, we assumed a difference in the recurrence rate of 20%. With a 0.05 setting (one-sided), power of 0.80 in a ×2 analysis, and an anticipated loss to follow-up of 20%, we intended to include 30 patients. Because of the low recruitment rate, the actual sample size was 23, which, yielded powers of 0.67 (one-sided) and 0.56 (two-sided).

## 3. Results

From the 23 patients included in the study, eight (34.8%) patients underwent tenotomies in both feet, and 15 (65.2%) underwent tenotomies in only one foot. A total of 99 tenotomy procedures were performed, all as prophylactic procedures. Baseline data on demographic characteristics and diabetes complications are shown in Table 1.

Regarding demographic characteristics and diabetes complications, we found that patients who underwent several tenotomies had more prevalence of hypertension and higher BMI values than isolated tenotomies patients (Table 1).

A total of 31 feet were analyzed, of which 11 feet (35.5%) underwent isolated tenotomies, and 20 (64.2%) underwent several tenotomies. All the post-operative cures healed in a median time of 10 IQR days (range, 7–14 days). During the follow-up period, no feet developed a recurrence event; in addition, eight (25.8%) feet developed a reulceration event in the following locations: five (62.5%) patients developed a transfer lesion to the third toe after tenotomy of the second toe, one (12.5%) patient developed a transfer lesion to the second toe after tenotomy of the first toe, and two (25%) patients developed a transfer lesion to the fourth toe after tenotomy of the third toe. Feet characteristics are shown in Table 2.

### Main Outcome

Patients with isolated tenotomies (*n =* 11, 35.48%) showed a higher rate of reulceration (*n =* 8, 72.7%) secondary to transfer lesions in the adjacent toes in a median time of nine and half weeks (IQR, 8–10.75 weeks). In addition, in the adjacent toes, we found more prevalence of hyperquerathosis (*n =* 11, 100%) in a median time of five weeks (IQR, 4–6 weeks) and minor lesions (*n =* 9, 81%) in a median time of six and a half weeks (IQR, 6–7.75 weeks). Adjacent claw toes were found in the group of patients who underwent isolated tenotomies (*n =* 11, 100%) in a median time of five weeks (IQR, 4.25–6 weeks). Barefoot peak pressure (1.48 ± 0.26 N/cm^2^) and integral pressure time (0.83 ± 0.11 N/cm^2^/s) in the non-tenotomy toes were higher in the group of patients who underwent isolated tenotomies. Finally, in the group of patients who underwent several tenotomies (*n =* 20, 64.52%), we found a higher rate of floating toes (*n =* 16, 80%) in comparison with isolated tenotomies patients in a median time of four weeks (IQR, 2–4) (Table 3).

After adjusting the results by the number of procedures performed, several tenotomies (patients who underwent three, four, and five procedures) showed to be more effective in reducing the reulceration rate in comparison with one tenotomy patients (Table 4).

## 4. Discussion

We found that patients who underwent several tenotomies had better clinical outcomes, such as lower reulceration rates, fewer minor lesions, less hyperkeratosis, and fewer claw toes in the adjacent toes. In addition, patients who underwent several tenotomies resulted in lower barefoot pressures beneath the hallux and minor toes, resulting in reducing the risk of these areas developing a new ulcer event. The data from this research support that patients with previous toe ulcers or thick calluses in the tip of the toe should undergo percutaneous flexor tenotomies in all the toes to reduce long-term complications.

Previous studies have analyzed transfer lesions that occurred after flexor tenotomies: Rasmussen et al. [12] found two transfer lesions (7%) 20 to 28 weeks after surgery, Tamir et al. [15] found nine transfer lesions (9%) eight weeks after surgery, and Van Netten et al. [11] found eight transfer lesions (21%). It is unfortunate that this latter cited study did not report the time to develop a transfer lesion, and additionally, they did not evaluate if the number of tenotomies implemented could influence the clinical outcome. To our knowledge, no prior study has evaluated if reulceration after flexor tenotomies can be related to isolated or several tenotomies in the affected foot. We found that eight feet (38.1%) of the sample developed a reulceration event and eight (72.7%) in the isolated tenotomies group, but no patients developed a recurrent event during the follow-up period. Regarding the time to suffer the reulceration event after tenotomies, Tamir et al. [15] found the time to reulcerate was eight weeks, similar results compared to ours, with a median time of nine and a half weeks in the isolated tenotomy group. However, Rasmussen et al. [12] found that patients who suffered from a reulceration event developed transfer lesions five and seven months after surgery, which is longer than in our results, which could be explained by Rasmussen et al. not following the patients on a monthly basis after healing, as IWGDF guidelines recommend. The only study that evaluated the location of the tenotomy and the reulceration found that after tenotomy of the hallux, two patients developed transfer lesions to the tip of the second toe, which is consistent with the results reported here [12].

Toe deformities after flexor tenotomies have only been evaluated in one prior study [11]. These authors found that one patient developed a dorsal ulcer secondary to dorsiflexion of the metatarsophalangeal joint, and in that respect, we found that all the patients subjected to isolated tenotomy developed a claw toe in the adjacent toe with no dorsal ulcer. The fact that our patients did not develop a dorsal ulcer might be related to the extra-depth therapeutic footwear.

To our knowledge, no previous research has evaluated the development of calluses and minor lesions in patients after prophylactic tenotomies; our research showed that patients who underwent isolated tenotomies suffered from high rates of calluses and minor lesions in comparison with the group of several tenotomies that did not show any calluses. In fact, this research is the first to evaluate barefoot pressures after tenotomy procedures, which could be linked to the appearance of calluses and minor lesions, as many authors have suggested [26]. Evaluating risk factors for plantar ulcers is an important factor, and further research in diabetic foot surgery should address this. Finally, floating toes were found in 80% of the subgroup of patients who underwent several tenotomies. Despite this, the patients did not develop any complications secondary to the hyperextension of the toe, and only one patient developed a distal ulcer secondary to a small shoe size that healed within the standard of care [28].

The present study includes the highest yet published sample of patients with flexor tenotomies on a prophylactic basis and a prospective design. This is the first study also, that analyzes clinical outcomes after flexor tenotomies divided by the number of tenotomies performed. Further research should focus on controlled studies, and additionally, the cost-effectiveness of the procedure should be investigated in comparison with other noninvasive interventions.

However, our results should be interpreted with caution because we only evaluated patients without ischemia, and patients with a poor vascular supply could benefit from this simple and non-aggressive procedure. Additionally, further research should confirm the effectiveness of percutaneous flexor tenotomies with alternative procedures such as toe orthosis and footwear modifications. Unfortunately, the sample size was reduced in number than originally calculated, due to the difficulty in recruiting patients and the difficulty in finding the requirement in characteristic needed.

## 5. Conclusions

Patients who underwent several tenotomies had better clinical outcomes after a 1-year follow-up period compared to isolated tenotomies. To reduce complications, clinicians should consider performing several tenotomies in patients with toe ulcers.

## Figures and Tables

**Figure 1 jcm-11-04093-f001:**
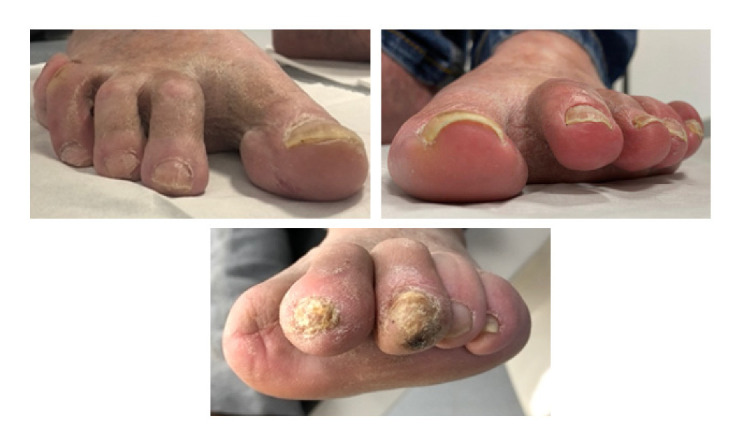
Toe deformities evaluated in the study. (**Left**), 2nd, 3rd, and 4th claw toe secondary to hallux flexor tenotomy; (**right**), 2nd, 3rd, 4th, and 5th floating toes secondary to several flexor tenotomies (all toes); (**down**), minor lesion in the tip of the 3rd toe.

**Table 1 jcm-11-04093-t001:** Differences between the risk factors for tenotomy groups (N *=* 23).

Baseline Characteristics	All Patients (N = 23)	Isolated Tenotomies Patients (*n* = 11)	Several Tenotomies Patients (*n* = 12)	*p*-Value [95% CI]
Male, *n* (%)	20 (87.0%)	10 (90.9%)	10 (83.3%)	0.590
Female *n*, (%)	3 (13.0%)	1 (9.1%)	2 (16.7%)	
Type 2 Diabetes, *n* (%)	22 (95.7%)	10 (90.9%)	12 (100%)	0.286
Type 1 Diabetes, *n* (%)	1 (4.3%)	1 (9.1%)	-	
Retinopathy, *n* (%)	12 (52.2%)	5 (45.5%)	7 (58.3%)	0.537
Nephropathy, *n* (%)	4 (17.4%)	2 (18.2%)	2 (16.7%)	0.924
Hypertension, *n* (%)	18 (78.3%)	6 (54.5%)	12 (100%)	**0.008 ***
Hypercholesterolemia, *n* (%)	22 (95.7%)	11 (100%)	11 (91.7%)	0.328
Cardiovascular disease, *n* (%)	11 (47.8%)	3 (27.3%)	8 (66.7%)	0.006
Neuropathy, *n* (%)	11 (100%)	11 (100%)	12 (100%)	-
Previous Ulceration, *n* (%)	31 (100%)	11 (100%)	12 (100%)	-
Permeable Pedal Pulses, *n* (%)	16 (69.6%)	8 (72.7%)	8 (66.7%)	0.752
Ankle Brachial Pressure Index, mean ± SD	1.18 ± 0.33	1.17 ± 0.28	1.19 ± 0.39	0.877
Toe Brachial Pressure Index, mean ± SD	0.76 ± 0.15	0.78 ± 0.13	0.75 ± 0.17	0.609
Transcutaneous Oxygen Pressure (mmHg), mean ± SD	30.08 ± 6.90	28.81 ± 7.09	31.25 ± 6.82	0.412
Mean age ± SD (years)	66.26 ± 11.20	68.9 ± 10.39	63.83 ± 11.81	0.288
Glycated hemoglobin mmol/mol (%), mean ± SD	7.54 ± 1.33	7.38 ± 1.34	7.69 ± 1.36	0.590
Diabetes mellitus (years), mean ± SD	14.69 ± 11.63	15.35 ± 15.35	13.08 ± 7.11	0.500
Body Mass Index (kg/cm^2^)	30.15 ± 3.73	28.36 ± 3.16	31.79 ± 3.56	**0.002 ***

SD, standard deviation; kg, kilograms; cm^2^, squared centimeters. * *p* < 0.05 indicates statistical significance.

**Table 2 jcm-11-04093-t002:** Feet characteristics (N *=* 31).

Feet Characteristics	Feet (N = 31)
Number of Tenotomies, *n* (%)	
One	11 (35.5%)
Two	0 (0%)
Three	4 (12.9%)
Four	4 (12.9%)
Five	12 (38.7%)
Ulcer Recurrence, *n* (%)	0 (0%)
Reulceration, *n* (%)	8 (25.8%)
Callus Formation, *n* (%)	11 (35.5%)
Minor Lesion, *n* (%)	9 (29%)
Adjacent Floating Toe, *n* (%)	17 (54.8%)
Adjacent Claw Toe, *n* (%)	12 (38.7%)
Peak Plantar Pressure beneath Hallux, (N/cm^2^)	0.26 ± 0.22
Pressure Time Integral beneath Hallux, (N/cm^2^/s)	0.09 ± 0.10
Peak Plantar Pressure beneath minor toes, (N/cm^2^)	0.57 ± 0.71
Pressure Time Integral beneath minor toes, (N/cm^2^/s)	0.31 ± 0.4

N, Newton; cm^2^, squared centimeters; s, seconds.

**Table 3 jcm-11-04093-t003:** Differences between the risk factors for tenotomy groups.

Baseline Characteristics	Feet with Isolated Tenotomies (*n* = 11)	Feet with Several Tenotomies (*n* = 20)	*p*-Value [95% CI]
Number of Tenotomies, *n* (%)			
One	11 (100%)	-	-
Two	-	-	-
Three	-	4 (20%)	-
Four	-	4 (20%)	-
Five	-	12 (60%)	-
Ulcer Recurrence, *n* (%)	-	-	-
Reulceration, *n* (%)	8 (72.7%)	0	**<0.001 ***
Callus Formation, *n* (%)	11 (100%)	0	**<0.001 ***
Minor Lesion, *n* (%)	9 (81.8%)	0	**<0.001 ***
Adjacent Floating Toe, *n* (%)	1 (9.1%)	16 (80%)	**<0.001 ***
Adjacent Claw Toe, *n* (%)	11 (100%)	1 (20%)	**<0.001 ***
Peak Plantar Pressure beneath Hallux, (N/cm^2^)	0.31 ± 0.18	0.14 ± 0.23	**0.04 ***
Pressure Time Integral beneath Hallux, (N/cm^2^/s)	0.13 ± 0.08	0.07 ± 0.14	0.09
Peak Plantar Pressure beneath minor toes, (N/cm^2^)	1.48 ± 0.26	0.07 ± 0.17	**<0.001 ***
Pressure Time Integral beneath minor toes, (N/cm^2^/s)	0.83 ± 0.11	0.03 ± 0.08	**<0.001 ***

N, Newton; cm^2^, squared centimeters; s, seconds. * *p* < 0.05 indicates statistical significance.

**Table 4 jcm-11-04093-t004:** Multiple comparison analyses between tenotomy groups.

	Number of Tenotomies Performed	Reulceration	*p*-Value [95% CI]
One-tenotomy reulceration patients (*n* = 8)	Three	0 (0%)	**<0.001 * [0.25–1.19]**
	Four	0 (0%)	**<0.001 * [0.25–1.19]**
	Five	0 (0%)	**<0.001 * [0.38–1.06]**
Three-tenotomies reulceration patients (*n* = 0)	One	8 (72.7%)	**<0.001 * [−1.19–0.25]**
	Four	0 (0%)	1.00 [−0.57–0.57]
	Five	0 (0%)	1.00 [−0.46–0.46]
Four-tenotomies reulceration patients (*n* = 0)	One	8 (72.7%)	**<0.001 * [−1.19–0.25]**
	Three	0 (0%)	1.00 [−0.57–0.57]
	Five	0 (0%)	1.00 [−0.46–0.46]
Five-tenotomies reulceration patients (*n* = 0)	One	8 (72.7%)	**<0.001 * [−1.06–0.38]**
	Three	0 (0%)	1.00 [−0.46–0.46]
	Four	(0%)	1.00 [−0.46–0.46]

** p* < 0.05 indicates statistical significance.

## Data Availability

Not applicable.

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
