# Peer review of "Safety and Efficacy of Several Versus Isolated Prophylactic Flexor Tenotomies in Diabetes Patients: A 1-Year Prospective Study"

_jcm, 2022, doi:10.3390/jcm11144093_

Round 1

Reviewer 1 Report

The main purpose of this paper is to assess long-term clinical outcomes of patients who underwent isolated versus several percutaneous flexor tenotomies for the treatment of toe deformities and previous diabetic foot ulcers. In my opinion this paper suffers from a couple of major biases :

1. The aim of this study is poorly intelligible: even if Authors present it as a prospective intervention trial it seems like a retrospective trial describing the prognosis at one year of a single procedure: percutaneous flexor tenotomies. We are given scares information on how patients were recruited, whether this procedure was compared with possible alternative procedure and additionally what the reason to perform an isolated as compared to several percutaneous  tenotomies.

2. Besides these points, moreover, the study seems to be underpowered with some groups containing lass than 10 patients: any statistical analysis seems consequently to be unaffordable, and no adjustment was done due to multiple comparisons analyses, increasing the risk of type I errors.

In conclusion, it seems that the primary goal of the study as stated by Authors in the introduction seems to be to properly analyze the outcomes after percutaneous flexor tendon tenotomies (both isolated or several), and, for this purpose they should be compared with other currently used procedures. In this context, furthermore one should perform a correct analysis of statistical power, seemingly increasing the number of observations.

Finally a copy-editing of English should be performed throughout the text (for instance  weak instead of weak at line 45).

Author Response

REVIEWER 1

Dear reviewer 1, thank you very much for your kind revision of our manuscript, we hope that all the changes recommended will improve the quality of the paper.

The main purpose of this paper is to assess long-term clinical outcomes of patients who underwent isolated versus several percutaneous flexor tenotomies for the treatment of toe deformities and previous diabetic foot ulcers. In my opinion this paper suffers from a couple of major biases:

  1. The aim of this study is poorly intelligible: even if Authors present it as a prospective intervention trial it seems like a retrospective trial describing the prognosis at one year of a single procedure: percutaneous flexor tenotomies. We are given scares information on how patients were recruited, whether this procedure was compared with possible alternative procedure and additionally what the reason to perform an isolated as compared to several percutaneous  tenotomies.

Dear reviewer 1, thank you very much for the comment, we added to the methods section a more precise explanation of patient’s recruitment: “All the patients included in the study suffered from a previous orthopedic treatment failure, defined as ulcer recurrence in the same toe by using toe spacers and therapeutic footwear modifications [9].” Lines 83-85.

Unfortunately, the current research does not compare percutaneous flexor tenotomies with other alternative procedures, we will add this issue in the discussion as a limitation: “Additionally, further research should confirm the effectiveness of percutaneous flexor tenotomies with alternative procedures such as toe orthosis and footwear modifications.” Lines 330-332.

Finally, the reason why we performed an isolated compared to several tenotomies is explained as follow: “Patients were subjected to isolated tenotomies when only one toe was affected with a previous toe ulcer, several tenotomies were performed in patients with one or more toes affected with previous toe ulcers, and in addition, any of the remaining toes were affected with a flexible toe deformity.” Lines 142-145.

  1. Besides these points, moreover, the study seems to be underpowered with some groups containing lass than 10 patients: any statistical analysis seems consequently to be unaffordable, and no adjustment was done due to multiple comparisons analyses, increasing the risk of type I errors.

As suggested by the reviewer, we added in the statistical analyses section the sample size calculation and the sample size power due to the low recruitment rate: “Sample size calculation was based on a 28-month follow-up study of patients with previous flexor tenotomy of the flexor digitorium longus to heal tip toe ulcers [10] showed a recurrence rate of 12.1%. As a relevant risk reduction, we assumed a difference in the recurrence rate of 20%. With 0.05 setting (one-sided), power of 0.80 in a ×2 analysis, and an anticipated loss to follow-up of 20%, we intended to include 30 patients. Because of the low recruitment rate, the actual sample size was 23, which, yielded powers of 0.67 (one-sided) and 0.56 (two-sided).” Lines 211-216

Finally, we explained the low recruitment rate in the discussion as a limitation: “Unfortunately, the sample size was reduced in number than previously calculated, due to the difficulty in recruiting patients and the difficulty in finding the requirement in characteristic needed.” Lines 332-334.

In conclusion, it seems that the primary goal of the study as stated by Authors in the introduction seems to be to properly analyze the outcomes after percutaneous flexor tendon tenotomies (both isolated or several), and, for this purpose they should be compared with other currently used procedures. In this context, furthermore one should perform a correct analysis of statistical power, seemingly increasing the number of observations.

As the reviewer pointed in the first comment, the main aim of this study was to compare the reulceration rate of tip toe ulcers after isolated vs several tenotomies. We did not analyze further techniques in comparison to flexor tenotomies and we explained this fact as a limitation in the discussion. We will analyze in further research what the reviewer recommends with other currently used procedures. Lines 330-332.

Finally a copy-editing of English should be performed throughout the text (for instance  weak instead of weak at line 45).

Dear reviewer the paper was previously sent to an English professional copy-editing service, we have sent the paper once again to solve grammar mistakes. We attach the English Editing Certificate.

Reviewer 2 Report

Reviewer Comments

Thank you very much for the opportunity to review the manuscript submission entitled: Safety and efficacy of several versus isolated prophylactic flexor tenotomies: A 1-year prospective study.

The current paper aims to assess long-term clinical outcomes of patients who underwent isolated versus several percutaneous flexor tenotomies for the treatment of toe deformities and previous diabetic foot ulcers. The data is interesting, and it has a relevant rationale; however, some limitations and constructive comments are pointed out below:

Specific comments

Title and Abstract

·      Include subjects such as diabetics in the title.

·      Include the mean age of the study population in the abstract.

·      Include the units of measurement for outcome measures.

·      Include MeSH terms as keywords.

Introduction

·      The scientific background and rationale for the investigation need to be emphasized. 

·      Use current literature to support your statements.

·      The hypothesis of the study needs to be stated.

Methods

·      Describe the setting, locations, and relevant dates, including periods of recruitment, exposure, follow-up, and data collection.

·      Needs more emphasis on Inclusion and exclusion criteria.

·      Explain how the study size was arrived at?

Statistical methods and results

·      What is the basis for the selection of statistical tests? Did the data (study variables) follow normal distribution?

·      Remove the zero before the decimal point when reporting a P-value. Do this in tables and main text wherever you reported a p-value.

·       

Discussion

·      Give a cautious overall interpretation of results considering objectives, limitations, the multiplicity of analyses, and results from similar studies.

·      Discuss the generalizability (external validity) of the study results.

Author Response

REVIEWER 2

Thank you very much for the opportunity to review the manuscript submission entitled: Safety and efficacy of several versus isolated prophylactic flexor tenotomies: A 1-year prospective study.

The current paper aims to assess long-term clinical outcomes of patients who underwent isolated versus several percutaneous flexor tenotomies for the treatment of toe deformities and previous diabetic foot ulcers. The data is interesting, and it has a relevant rationale; however, some limitations and constructive comments are pointed out below:

Dear reviewer 2, we appreciate your kind revision of our manuscript very much, we hope that all the changes and recommendations will improve the quality of the paper.

Specific comments

Title and Abstract

  • Include subjects such as diabetics in the title.

Dear reviewer following your recommendation, we changed the title to: “Safety and efficacy of several versus isolated prophylactic flexor tenotomies in diabetes patients: A 1-year prospective study”

  • Include the mean age of the study population in the abstract.

Mean age of the study population was included in the abstract accordingly: “Twenty-three patients (mean age 66.26 ± 11.20, years) who underwent prophylactic percutaneous flexor tenotomies secondary to tip-toe ulcers participated in this 1-year prospective study”. Line15.

  • Include the units of measurement for outcome measures.

      Units of measurement for outcome measures were included in the abstract accordingly: “peak plantar pressure (PPP – N/cm2) and pressure/time Integral (PTI – N/cm2/s) in the hallux and minor toes after a 1-year follow-up period”. Line 20.

  • Include MeSH terms as keywords.

      Keywords were changed to MeSH terms only accordingly (diabetic foot; surgery, deformity, prevention). Line 29.

Introduction

  • The scientific background and rationale for the investigation need to be emphasized.      

Scientific background was restructured accordingly, additionally the importance of the rationale was pointed out: “Flexor tendon tenotomies are widely used to treat and prevent toe ulcer, despite this, no previous research has analyzed outcomes after percutaneous flexor tendon tenotomies and if there exists any relationship between patients with several versus isolated tenotomies.” Lines 70-73.

  • Use current literature to support your statements.  

Thank you very much for your comment, our statements have been based on the last update of the International Working Group Diabetic Foot (IWGDF) guidance (2019 update, 2023 is coming), additionally, flexor tendon tenotomies statements are based on the few published papers on the literature about the topic:

- Kearney TP, Hunt NA, Lavery LA. Safety and effectiveness of flexor tenotomies to heal toe ulcers in persons with diabetes. Diabetes Res Clin Pract. 2010;89(3):224-226;

- van Netten JJ, Bril A, van Baal JG. The effect of flexor tenotomy on healing and prevention of neuropathic diabetic foot ulcers on the distal end of the toe. J Foot Ankle Res. 2013;6(1):3.

- Rasmussen A, Bjerre-Christensen U, Almdal TP, Holstein P. Percutaneous flexor tenotomy for preventing and treating toe ulcers in people with diabetes mellitus. J Tissue Viability. 2013;22(3):68-73;

- Tamir E, McLaren AM, Gadgil A, Daniels TR. Outpatient percutaneous flexor tenotomies for management of diabetic claw toe deformities with ulcers: a preliminary report. Can J Surg. 2008;51(1):41-44.))

  • The hypothesis of the study needs to be stated.

      Dear reviewer, thank you very much for the comment, the hypothesis of the study was added: “We hypothesize that, to implement several percutaneous flexor tenotomies in patients with previous tip-toe ulcers can reduce the reulceration rate in comparison with isolated percutaneous flexor tenotomies.” Lines 73-75.

Methods

  • Describe the setting, locations, and relevant dates, including periods of recruitment, exposure, follow-up, and data collection.

      Patients were included in the research from April 2019 to January 2021, we did not added location to the manuscript due to Journal policies (blinding), all the procedures were performed in a specialized diabetic foot unit. Surgical techniques were carried out by the same surgeon in the same operating room (“All the procedures were performed by the same senior surgeon (JLLM) in the same operating theatre.”). Lines 135-136.

Additionally, we included that all the clinical and biomechanical evaluations were carried out in a specialized biomechanics lab (“All clinical evaluations were carried out in a specialized biomechanics laboratory.”). Lines 166-167.

Data collection was specified in the clinical evaluation (“At baseline, clinical characteristics were collected in the case report form (crf) after the patient signed informed consent on day zero.”). Line 111.

  • Needs more emphasis on Inclusion and exclusion criteria.

      Inclusion and exclusion criteria were emphasized according to the recommendation of the reviewer: “The inclusion criteria were confirmed type 1 or type 2 diabetes, age > 18 years, affected with flexible toe deformities, history of tip-toe DFU, and loss of protective foot sensation because of peripheral neuropathy (DPN).

Exclusion criteria were ulcers during the examination, transmetatarsal or major amputation in the contralateral limb (below or above the knee), history of rheumatoid disease, other causes of neuropathy, critical limb ischemia as defined according to the IWGDF guidance [16], and the need for walking aids. Patients with previous toe surgery were also excluded (both musculoskeletal and soft tissue procedures).” Lines 86-93.

  • Explain how the study size was arrived at?         

Sample Size calculation was included in the statistical analyses section: “Sample size calculation was based on a 28-month follow-up study of patients with previous flexor tenotomy of the flexor digitorium longus to heal tip toe ulcers [10] showed a recurrence rate of 12.1%. As a relevant risk reduction, we assumed a difference in the recurrence rate of 20%. With 0.05 setting (one-sided), power of 0.80 in a ×2 analysis, and an anticipated loss to follow-up of 20%, we intended to include 30 patients.” Due to the low recruitment rate, we calculated the sample size power: “Because of the low recruitment rate, the actual sample size was 23, which, yielded powers of 0.67 (one-sided) and 0.56 (two-sided).” Lines 211-217.

Statistical methods and results

  • What is the basis for the selection of statistical tests? Did the data (study variables) follow normal distribution?

      The distribution of the study variables followed normal distribution after Kolmogorov-Smirnov test was used: “The assumption of the normality of all continuous variables was verified using the Kolmogorov-Smirnov test. Quantitative variables were presented as the mean and standard deviation (SD), while qualitative variables were presented as percentages and frequencies. To explore differences in clinical features between patients with several versus isolated tenotomies, the Chi-square test for categorical variables and the Student’s t-test for quantitative variables were performed.” Lines 201-206. 

  • Remove the zero before the decimal point when reporting a P-value. Do this in tables and main text wherever you reported a p-value.           

Thank you very much for the appreciation, zero before the decimal point when reporting P-value were removed along the text, included in tables and main text.       

Discussion

  • Give a cautious overall interpretation of results considering objectives, limitations, the multiplicity of analyses, and results from similar studies.

      We have tried to give a cautious overall interpretation of results considering all the issued mentioned by the reviewer: “We found that patients who underwent several tenotomies had better clinical out-comes, such as lower reulceration rates, fewer minor lesions, less hyperkeratosis, and fewer claw toes in the adjacent toes. In addition, patients who underwent several tenotomies resulted in lower barefoot pressures beneath the hallux and minor toes, resulting in reducing the risk of these areas developing a new ulcer event.”. Lines 269-273.

  • Discuss the generalizability (external validity) of the study results.          

The generalizability of the study results is showed as follow in the discussion section: “The data from this research support that patients with previous toe ulcers or thick calluses in the tip of the toe should undergo percutaneous flexor tenotomies in all the toes to reduce long-term complications.” Lines 273-276.

And “To reduce complications, clinicians should consider performing several tenotomies in patients with toe ulcers.” Lines 338-340.

Round 2

Reviewer 1 Report

Dear Authors:

My main concern about this study remains: also also admitted by Authors it is largely underpowered, far from reaching a one-sided power of .80. Furthermore no adjustement for multiple comparisons has been done. When designing a prospective study, the power is usually calculated prior to patients  recruitment: all this confirms that it is a  retrospective review (not a prospective study as from the title) of this procedure and, anywhere it should be clearly stated that the study has not a prospective design and that it is underpowered, being therefore, not able to give definitive clear asnswers but only suggestions

Author Response

RESPOND TO REVIEWERS

REVIEWER 1

My main concern about this study remains: also also admitted by Authors it is largely underpowered, far from reaching a one-sided power of .80. Furthermore no adjustement for multiple comparisons has been done. When designing a prospective study, the power is usually calculated prior to patients  recruitment: all this confirms that it is a  retrospective review (not a prospective study as from the title) of this procedure and, anywhere it should be clearly stated that the study has not a prospective design and that it is underpowered, being therefore, not able to give definitive clear asnswers but only suggestions

Dear reviewer, thank you very much once again for the revision of our manuscript.

Regarding your main concern about our manuscript, we agree with the one-sided low power of the sample, we have properly discussed it as a limitation in the discussion section: “Unfortunately, the sample size was reduced in number than previously calculated, due to the difficulty in recruiting patients and the difficulty in finding the requirement in characteristic needed.” Lines 300-330.

Regarding your statement of the retrospective nature of the analyses, we do not really agree with it, patients were operated and were prospectively (from April 2019) followed-up for 1 year (last study visit was performed in January 2021 for the last patient), the total follow-up period was 21 months, in that long time we had problems in the recruitment, and we had to stop it before reaching the final sample size calculation.

It is true that some groups of tenotomies have lower sample size than others, for that reason and according to the recommendations of the reviewer we have added the adjusted analyses with multiple comparisons, and still, after it, the results of the several tenotomies groups are better in terms of reduction of reulceration rate. We added these new analyses to the last part of the results: “After adjusting the results by the number of procedures performed, several tenotomies (patients who underwent three, four and five procedures) showed to be more effective in reducing the reulceration rate in comparison with one tenotomy patients (Table 4).

Table 4. Multiple comparison analyses between tenotomy groups.

Number of tenotomies performed

Reulceration

P-value

[95% CI]

One-tenotomy reulceration patients

(n=8)

Three

0 (0%)

<.001* [0.25 – 1.19]

Four

0 (0%)

<.001* [0.25 – 1.19]

Five

0 (0%)

<.001* [0.38 – 1.06]

Three-tenotomies reulceration patients

(n=0)

One

8 (72.7%)

<.001* [-1.19 – 0.25]

Four

0 (0%)

1.00 [- 0.57 – 0.57]

Five

0 (0%)

1.00 [- 0.46 – 0.46]

Four-tenotomies reulceration patients

(n=0)

One

8 (72.7%)

<.001* [-1.19 – 0.25]

Three

0 (0%)

1.00 [- 0.57 – 0.57]

Five

0 (0%)

1.00 [- 0.46 – 0.46]

Five-tenotomies reulceration patients

(n=0)

One

8 (72.7%)

<.001* [- 1.06 – - 0.38]

Three

0 (0%)

1.00 [- 0.46 – 0.46]

Four

0 (0%)

1.00 [- 0.46 – 0.46]

. *P < .05 indicates statistical significance.

Lines 241 - 246.

Additionally, we explained it in the statistical analyses section: “A secondary multiple comparison sub analyses (Bonferroni test) was performed between the subgroups of tenotomy patients (one tenotomy, three tenotomies, four tenotomies and five tenotomies) for the primary outcome measure.” Lines 192-195.

Reviewer 2 Report

The authors have answered all of my questions and the paper has been greatly improved. 

Author Response

Thank you very much for your comments.